# Metallurgical Soldering of Duplex CrN Coating in Contact with Aluminum Alloy

**Pal Terek** [1], **Lazar Kovačević** [1], **Aleksandar Miletić** [1,2], **Branko Škorić** [1,*], **Janez Kovač** [3] and **Aljaž Drnovšek** [3]

1    Faculty of Technical Sciences, University of Novi Sad, Trg Dositeja Obradovića 6, 21000 Novi Sad, Serbia; palterek@uns.ac.rs (P.T.); lazarkov@uns.ac.rs (L.K.); aleksandar.miletic@polymtl.ca (A.M.)
2    Department of Engineering Physics, Polytechnique Montreal, Montreal, QC H3T 1J4, Canada
3    Jožef Stefan Institute, Jamova 39, 1000 Ljubljana, Slovenia; janez.kovac@ijs.si (J.K.); aljaz.drnovsek@ijs.si (A.D.)
*    Correspondence: skoricb@uns.ac.rs; Tel.: +381-21-485-2342

**Abstract:** Coatings deposited by physical vapor deposition (PVD) significantly reduce the wear of high pressure die casting tools; however, cast alloy soldering still has a strong negative effect on production efficiency. Although a lot of research has been already done in this field, the fundamental understanding of aluminum alloy soldering toward PVD coatings is still scarce. Therefore, in this work the performance of CrN duplex coatings with different roughness is evaluated by a modified ejection test performed with delayed (DS) and conventional casting solidification (CS). After the ejection tests, sample surfaces and layers were subjected to comprehensive characterizations of their morphological and chemical characteristics. Considerably lower values of the ejection force were recorded in DS experiments than in CS experiments. Surface roughness played an important role in the CS experiments, while samples with different surface topographies in the DS experiments performed in a similar fashion. The decrease in the ejection force, observed in DS tests, is attributed to the formation of a thick Cr–O layer on CrN coating which reduced soldering and sliding friction against thick Al–O casting scale. The Cr–O layer formed in DS experiments suffered from diffusion wear by cast alloy. The observed oxidation phenomena of nitride coatings may be utilized in a design of non-sticking coatings.

**Keywords:** die casting; aluminum; tool life; soldering; ejection test; CrN; surface roughness; oxidation; diffusion wear

## 1. Introduction

High pressure die casting (HPDC) is a technology used for the mass production of near-net shape parts of non-ferrous alloys, with thin walls and smooth surfaces. Due to the ever-increasing application of lightweight components in automotive products and other products, the use of HPDC technology for the production of aluminum alloy castings is constantly expanding. This kind of large volume production is economically justified only by highly efficient production of high-quality parts.

During operation die (tool) surfaces are exposed to wear by: erosion, corrosion and soldering, thermal cycling fatigue [1,2], and adhesion [3]. These processes affect the tool life and casting quality, but more importantly they increase the production costs due to increased: machine down times, number of rejected castings [3,4], energy and materials consumption. Aluminum alloys have the highest affinity toward iron contained in die steel materials and consequently induce the most pronounced soldering. To avoid cast alloy soldering before every casting cycle die-lubricant is sprayed on tool surfaces. The demands in terms of casting surface quality have been constantly increasing. In order to

achieve this high surface quality, specific casting processing conditions, which significantly increase tool wear due to the increased erosion and soldering effects, are required.

A prominent approach in the reduction of tool wear and improvement of tool performance is the application of thin ceramic coatings which are deposited by physical vapor deposition (PVD) techniques [1–3,5]. When ceramic coating is applied, the tool erosion is suppressed due to coating high hot hardness; corrosion and soldering are reduced due to high inertness and thermal stability of ceramic coating materials [1,6,7]; and finally, resistance to thermal fatigue cracking is improved by applying coatings of increased toughness, such as coatings of nanocomposite and nanolayer design [1,5,8]. In order to attain all these properties in one coating, the system of coating layers have to be adequately designed, as proposed by Lin et al. in [1]. Owing to its relatively high hardness, high oxidation and corrosion resistance, thermal stability [9] and low stress [10], CrN is still one of the commercially most used coatings for protection of tools HPDC of aluminum alloys. However, in recent years chromium-based coatings such as CrAlN [1,11], AlCrN [5,12], CrN/AlN [13], $Cr_2O_3$ [1,13], AlCrSiN [5,14] of different design received a considerable attention in protection of HPDC tools.

Nowadays, the improvement of casting release (ejection) from a die and reduction of lubricant consumption have been catching a significant attention in HPDC industry [12,15]. Additionally, the reduction of cast alloy soldering on long die-cores is still a great technological challenge. Therefore, in recent years, numerous studies have been focused on a topic of application of PVD coatings on HPDC tools [5,6,12,16]. Paiva et al. [5] evaluated the performance of AlCrN and two nanocomposite coatings ($AlTiN/Si_3N_4$; $AlCrN/Si_3N_4$) in high temperature tribological tests and in real HPDC industrial production. They showed that the tool life was considerably improved by application of nanocomposite coatings. Nunes et al. [16] investigated $Ti_xAl_{x-1}N$ coatings with different aluminum content, they compared the coating behavior obtained in tribological tests with the behavior obtained in HPDC industrial trials. They revealed that $Ti_xAl_{x-1}N$ coating with higher aluminum content provides better protection for HPDC tools. Bobzin et al. [6] employed a rotating immersion test (laboratory tests) and HPDC trials for evaluation of a $CrN/AlN/Al_2O_3$ coating system and two commercial coatings. $CrN/AlN/Al_2O_3$ coating system exhibited very good behavior in both kinds of tests, however, its top $Al_2O_3$ layer suffered from thermal cracking caused by phase change. Wang et al. [12] investigated the soldering performance of several nitride coatings using the aluminum adhesion test (laboratory test) and HPDC plant trails without application of die lubricants. Based on their findings, they proposed AlCrN coating as the most optimal candidate for the lubricant-free die casting. They measured the lowest force for separation from a casting and revealed the lowest soldering tendency for this coating.

Generally, the cast alloy soldering phenomena are divided into mechanical soldering (sticking) and metallurgical soldering [17]. Since many PVD coatings used in HPDC industry are inert to molten aluminum alloys [6,7], metallurgical soldering effects are easy to overlook and are generally not well recognized by scientific community. Formation of a built-up layer in the contact of a cast alloy and coated tool surfaces is usually attributed to mechanical soldering [3,18]. Nevertheless, evidences of metallurgical phenomena are present in literature. For example, corrosion of underlying substrate in a contact with molten aluminum alloy occurs through coating growth defects [19,20]. In such a process, a casting hooks the coating which hampers the casting ejection and consequently causes coating detachment. To prevent this, Abusuilik B. Saleh proposed intermediate coating treatment [19], while sealing of the coating defects by atomic layer deposition of thin $Al_2O_3$ layer [20] is also a promising approach. Besides soldering through coating defects, cast alloy can firmly bond in a form of a built-up layer to smooth coated surfaces which is often called sticking. Such remnants of the cast alloy on coated surfaces indicate hampered casting ejection. We believe that this process occurs due to the chemical compatibility or inter-diffusion between paired materials, and it should be classified as coatings metallurgical soldering. In few recent works [3,12,16] this phenomenon was observed, however, answers about fundamental mechanisms behind it were neither given nor discussed. Metallurgical soldering has not even been addressed for CrN coating which is the most investigated coating for protection of HPDC tools. Further development of soldering (sticking) resistant PVD coatings requires

thorough understanding of processes involved in interaction of the cast alloy with the coating. This was our primary motivation for performing the investigation presented herein.

Evaluation of performance of coating materials through practical experiments is time consuming, expensive, less controllable, and is highly limited in terms of studying of isolated, specific wear mechanisms (soldering, erosion, thermal fatigue) [21]. On the other hand, laboratory experiments are simple, quick, they provide isolation of a single wear mechanism, have high repeatability, and allow quantification of wear. One group of the laboratory tests used for evaluation of the soldering tendency, are separation tests [21]. These tests involve metal casting process for production of sample-casting assembly and a mechanical process of separation of a sample from a casting. In this way, most important processes that lead to cast alloy soldering and formation of a built-up layer (galling) are simulated. Therefore, these tests are considered the most appropriate for soldering evaluation.

A separation test used for evaluation of cylindrical pin samples is known as an ejection test. The improved ejection test, developed in our previous work [21], is simple to perform, however, it simulates only one casting cycle. In this test, the time cast alloy spends in contact with a pin sample is very short and therefore mainly mechanical soldering effects are simulated [21].

One approach to induce metallurgical soldering mechanisms is repetition of ejection tests in hundreds, or thousands of casting cycles which is highly impractical. The other, less time consuming and less expensive approach is extension of the time a molten metal stays in contact with coated surfaces. With the goal to enhance metallurgical soldering effects, and to have more comprehensive understanding of processes occurring in the contact of coated tool surfaces with molten aluminum alloys, in this investigation the ejection test was modified by adopting the latter approach, i.e., by introducing a delayed casting solidification. The performance of CrN duplex coatings in the contact with Al–Si–Cu alloy was studied. CrN was chosen as a model coating for this investigation because it is commonly applied in the field, it is simple for evaluation, and can help in understanding the soldering wear of other Cr-N-based coatings. CrN coating was prepared to different degrees of surface roughness. The obtained results are compared to results of our previous study in which conventional solidification methodology was used.

## 2. Materials and Methods

Soldering performance of duplex CrN coated core pins in contact with Al–Si–Cu alloy was evaluated by ejection test performed in two configurations. A standard configuration involved sting solidification, while the newly proposed configuration involved delayed casting solidification, performed with two periods of delay. Surface characterization of samples from both experiments is performed to explain the trends observed in quantitative data for both tests. In addition, coated samples were annealed in a separate experiment in order to have better understanding of heat-induced changes inside a CrN coating layer.

### 2.1. Samples Preparation

Substrates used in this investigation were produced from a quenched and double tempered EN X27CrMoV51 hot-working tool steel. Substrates were heated to 1000 °C and kept for 30 min for austenitizing, which was followed by oil quenching and double tempering (1 h at 620 °C and for 1 h at 500 °C). Substrates were machined by applying a sequence of procedures regularly used in production of HPDC tool parts, namely by turning, grinding and polishing. Samples were produced in two shapes, disc samples and cylindrical samples. Disc-shaped samples, with the dimensions $\phi 25 \times 5$ mm, were produced for characterization of materials properties and annealing experiment. Cylindrical pin-shaped samples, with the dimensions of the working part $\phi 15 \times 100$ mm, were used in ejection tests for evaluation of soldering tendency. For detailed drawing of pin samples please refer to our previous work [3]. Hardness obtained after quenching and double tempering of steel pin samples was $455 \pm 41$ HV$_{30}$.

CrN duplex composite layers were produced by plasma nitriding of steel substrates with subsequent deposition of CrN coating. Plasma nitriding was performed in a unit equipped with a pulsed plasma generator (ION-25I, IonTech, Sofia, Bulgaria). The nitriding process was performed in atmosphere with gas ratio of $H_2:N_2$ = 3:1 during a 12 h long processing cycle with 0.6 duty cycle.

In order to maximize adhesion of CrN layer a compound layer formed during nitriding was removed by polishing. The polishing procedure was performed using diamond paste with 3 μm and 6 μm granulations. As a result, two groups of samples (surfaces) of different roughness, were formed. Rough samples (R) were obtained by polishing with 6 μm paste, while smooth samples (S) were obtained by two-step polishing with 6 and 3 μm diamond paste.

Prior to coating deposition samples were degreased and cleaned in ultrasonic baths with deionized water and detergents and dried in hot air. CrN coating was deposited in an industrial thermionic arc ion plating system (BAI 730M, Balzers, Balzers, Liechtenstein) with the 2-fold samples rotation employed. More information on the employed deposition system might be found in [22]. The coating preparation process begun by heating samples in plasma to 450° which was followed by ion etching employing 200 V DC bias for 15 min. A thin Cr layer was deposited on substrates to improve CrN coating adhesion. During deposition, the bias voltage of −125 V was applied, while nitrogen partial pressure was kept constant. The typical deposition rate was around 50 nm/min. After the deposition, a group of smooth samples was submitted to additional polishing with 3 μm diamond paste. As a result, post deposition polished (PP) samples were obtained. Designation of all samples used in this study, procedures performed in their production and their roughness are presented in Table 1.

**Table 1.** Samples designations, production procedures and roughness parameters.

| Group of Samples | Rough | Smooth | Post Polished |
|---|---|---|---|
| Sample name | CrN-R | CrN-S | CrN-PP |
| Nitriding | × | × | × |
| Substrate polishing after nitriding 3 μm | × | × | × |
| Substrate polishing after nitriding 6 μm | | × | × |
| CrN coating | × | × | × |
| Polishing after coating deposition 6 μm | | | × |
| $R_a$ [μm] | 0.145 | 0.032 | 0.027 |
| $R_{sk}$ | −0.179 | 0.491 | −1.162 |

## 2.2. Soldering Evaluation–Ejection Test

The employed ejection test is explained in detail in our previous works [3,21]. In this test a cylindrical pin-shaped sample is cast-in in an Al–Si–Cu alloy casting. In this way a pin-casting assembly is obtained. In the next step, the pin sample is ejected from the casting. During the ejection process a force-displacement diagram is recorded, with the maximum force representing a quantitative measure of the bonding strength between paired materials (pin and casting materials), i.e., their soldering tendency.

In this study, pin-casting assemblies were produced in two configurations (methods). In the first configuration the casting was performed by gravity pouring molten EN AC-46200 aluminum alloy, at temperature of 730 °C, into a specially designed steel die [3,21], preheated to temperature of 320 °C. After the die was filled, the casting is allowed to solidify. In the rest of the article this configuration will be referred as conventional solidification method. The second configuration is a modification of the previous, it was performed using the same experimental die. Before casting, the die with mounted sample was preheated in a furnace for 40 min to achieve a target temperature of 600 °C. After cast metal was poured, the die was placed into a furnace heated to 700 °C to delay the casting solidification for a predetermined time (5 min and 20 min). After the predetermined time, the die was taken out of the furnace and the casting was allowed to solidify. This test configuration is called delayed solidification method and it is illustrated in Figure 1. This procedure was developed to intensify and extend corrosion processes that occur between a casting and a pin sample.

The conventional solidification experiments were conducted with three times repetitions while the delayed solidification experiments were performed without repetitions. Abbreviations used for designation of different test configurations are: CS—conventional solidification, DS 5—delayed solidification for 5 min and DS 20—delayed solidification for 20 min.

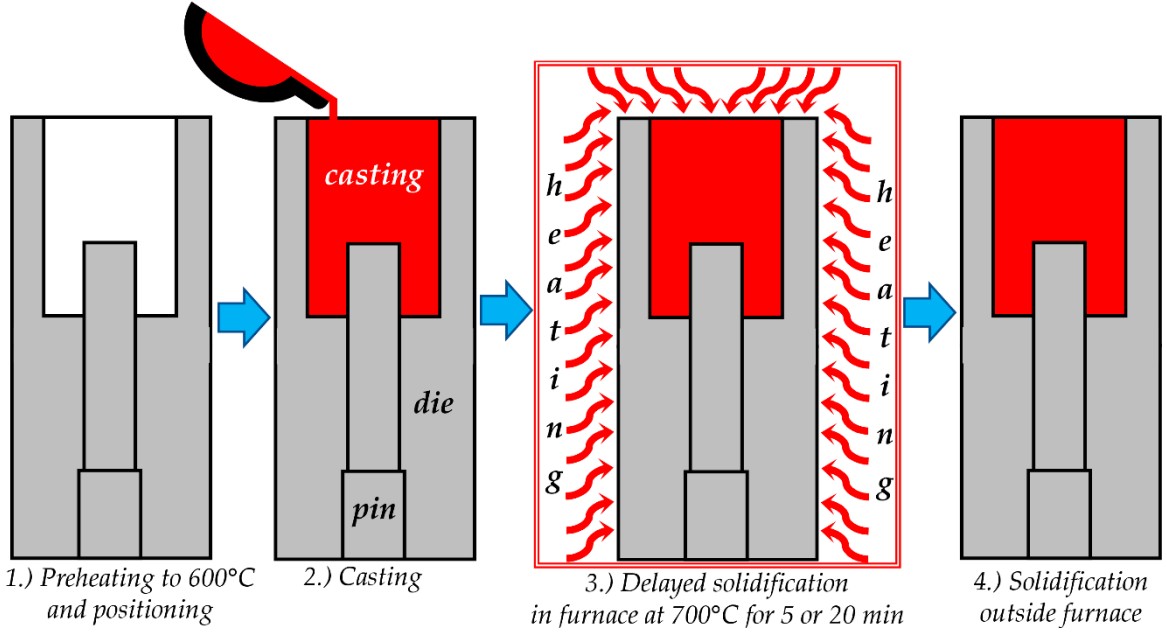

**Figure 1.** Schematic illustration of the casting method with the delayed solidification.

The temperature of die and sample surfaces were, for chosen conditions, determined in trial experiments by using exposed junction K-type thermocouples made from 0.25 mm wires (Omega Engineering Inc., Norwalk, Connecticut, USA) and by infrared thermometer. After taking the filled die out of the furnace, a casting cooling curve was recorded by immersing a K-type thermocouple casting's thermal axis.

A tensile testing machine (ZDM 5/91, VEB, Leipzig, Germany) was used for ejection of pins out of castings. More details can be found in our previous works [3,21]. During the ejection, a force-displacement curve was recorded (ejection curve). The force recorded during the test represents the soldering tendency of a cast alloy toward pin material. In the present study, the highest force recorded during the ejection test was chosen as a quantitative parameter for comparing the behavior of samples subjected to different experimental conditions.

*2.3. Annealing Experiment*

In order to evaluate the effect of high temperature on the structure and chemical composition of CrN duplex coating, annealing experiment was conducted. Disc shaped sample was placed in a laboratory tubular furnace preheated to 650 °C and kept there for 75 min in ambient air. The annealing time was chosen to equal the total time samples spend at high temperature in the 20 min delayed solidification test. In that test, samples are preheated for 40 min, solidification is delayed for 20 min, casting solidification lasted 15 min, which in total gives period of 75 min. The temperature of 650 °C was chosen as an average temperature to which the samples were exposed in DS 20 test.

*2.4. Samples Characterization*

Surface roughness was acquired by a stylus profilometer (Talysurf, Taylor Hobson, Leicester, United Kingdom). Instrumented hardness tester (H100C, Fischerscope, Windsor, CT, USA) was used for the determination of mechanical properties of nitrided layer and CrN coating. During the indentation

tests, the load of 50 mN and 100 mN was applied. A hardness profile was acquired by making indentations with loads of 100 mN along the thickness of the plasma nitrided layer. This hardness depth profile was used to determine the nitrided case thickness, in compliance with EN ISO 2639:2002, and the maximal layer hardness. After the ejection tests, examination of sample surfaces and cross sections provided additional information about the material behavior and about the soldering processes. Focused ion beam (FIB) (Helios Nanolab 650i, Fei, Hillsboro, Oregon, OR, USA) equipped with energy dispersive spectroscopy (EDS) was used for sample surface and cross-sectional analyses. In order to precisely identify the composition of very thin layers, samples were subjected to time of flight secondary ion mass spectroscopy (ToF-SIMS). ToF–SIMS instrument (ToF–SIMS 5, IONTOF, GmbH, Münster, Germany) equipped with a Bi liquid metal ion gun with a kinetic energy of 30 keV was used. The SIMS spectra were measured by scanning Bi+ ion beam over an area of $100 \times 100$ μm in size. SIMS depth profiles were measured in a dual beam depth profiling mode using a 2 keV $Cs^+$ ion beam rastering over an area of $0.4 \times 0.4$ mm for sputtering. Etching rate was estimated to be 0.20 nm/s.

## 3. Results

### 3.1. Materials and Layers Properties

Plasma nitriding process resulted in $90 \pm 10$ μm thick nitriding layer, which consisted of 87 μm thick diffusion layer and 3 μm thick compound layer. Hardness of the nitrided layer after the compound layer was removed was $1300 \pm 75$ $HV_{0.01}$.

Figure 2 presents results of the cross-sectional FIB and EDS analysis of CrN coating in its initial state. CrN coating was $2.7 \pm 0.25$ μm thick and exhibited hardness of $2735 \pm 235$ $HV_{0.05}$. The FIB cross-sectional analysis revealed a fine-grained compact microstructure of CrN single-layer coating, Figure 2a. Grains of relatively even size are uniformly distributed in the whole coating layer. According to EDS line analysis all chemical elements are uniformly distributed across the coating layer, Figure 2b. EDS analysis shows CrN coating is under-stoichiometric which is in agreement with results published by our team members in [23,24], where CrN coating was produced in the same way and where it was shown CrN coating was of sub-stoichiometric $Cr_2N$ composition. Considering that values in EDS analysis are presented in number of counts these results should be used only for qualitative evaluations and comparisons. Roughness of samples from different groups are presented in Table 1. The highest average roughness ($R_a$) was measured on samples from the rough group. Low and similar average roughness was measured on samples from the smooth and post polished groups. Although later two groups have been characterized by similar average roughness, skewness roughness parameter ($R_{sk}$) clearly indicates that surface topography of samples from these groups differ significantly.

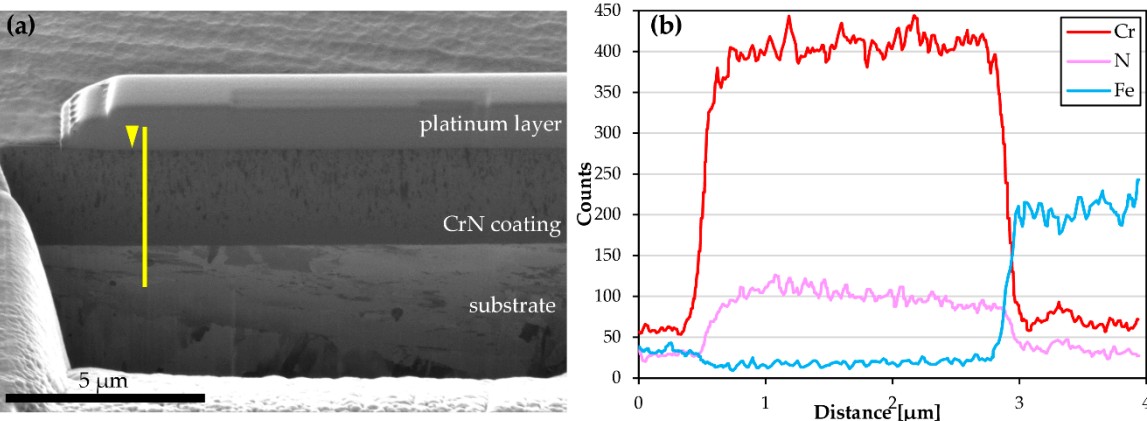

**Figure 2.** CrN coating in initial state, (**a**) ion induced secondary electron image of CrN coating cross-section, (**b**) chemical composition obtained by EDS analysis along the line depicted in the image (**a**).

## 3.2. Results of the Ejection Tests

Values of the maximal ejection force, recorded in all ejection tests, are jointly presented with the surface roughness ($R_a$) in Figure 3. Several trends are observable in the presented graph. First, the average ejection force in CS experiment increases with the decreased roughness. This was reported in our previous study [3]. Second, the ejection force in DS experiments was considerably lower than in CS experiments. Third, the longer the delay in solidification the lower the ejection force. Fourth, similar ejection force was measured in DS experiments on samples of different roughness. Altogether, the ejection force in DS experiments is reduced when compared to CS experiment, and there is a clear trend between the level of reduction and surface roughness, with the highest reduction observed for the post-polished sample. For better comprehension, the level of reduction of the ejection force is presented in percentages in Table 2.

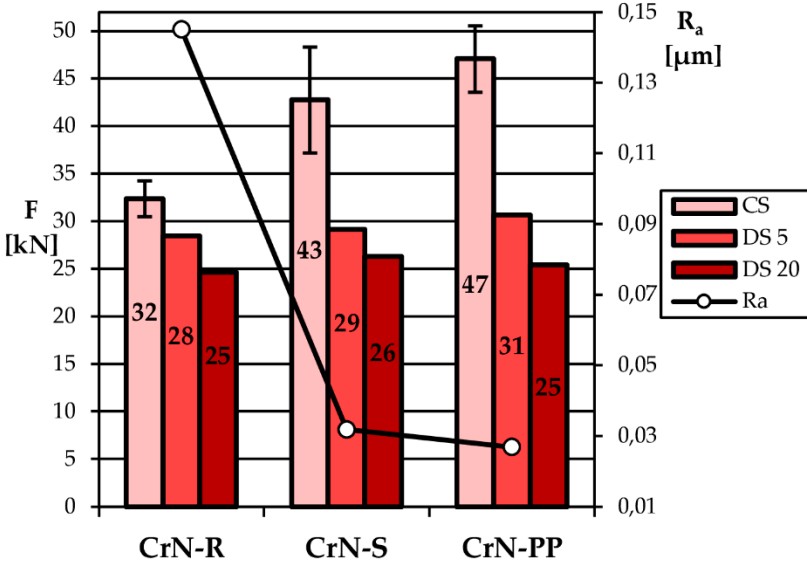

**Figure 3.** Maximal ejection force obtained for CrN pins with the different roughness in different experimental setups, error bars represent the ±1, 95% confidence interval (CI).

**Table 2.** Percentage (%) of the reduction in ejection force in delayed casting solidification (DS) experiments in comparison to values obtained in CS experiments.

| Sample Name | CrN-R | CrN-S | CrN-PP |
|:---:|:---:|:---:|:---:|
| **DS 5** | 12.5 | 32.5 | 34 |
| **DS 20** | 22 | 39.5 | 47 |

## 3.3. Cross Sectional Analysis of Coated Samples After Ejection Tests

Results of FIB analysis performed on CrN-R sample subjected to CS experiment are presented in Figure 4. Cross sectional analysis was performed on a location with a cast alloy built-up layer present in one typical micro groove (Figure 4a,b). This location was in a contact with the cast alloy with part of it left on the coating surface after the ejection process. The cross-sectional image indicates that CrN layer stayed intact. The layer was not damaged by tribological (mechanical) processes and its initial fine-grained microstructure was not changed (Figure 4b). Contrast observed in the built-up layer (Figure 4b) shows that different Al–Si–Cu cast alloy phases are present in the layer. Such a contrast in images formed by ion induced secondary electrons arises due to differences in ion channeling in phases with diverse crystalline structures and orientations.

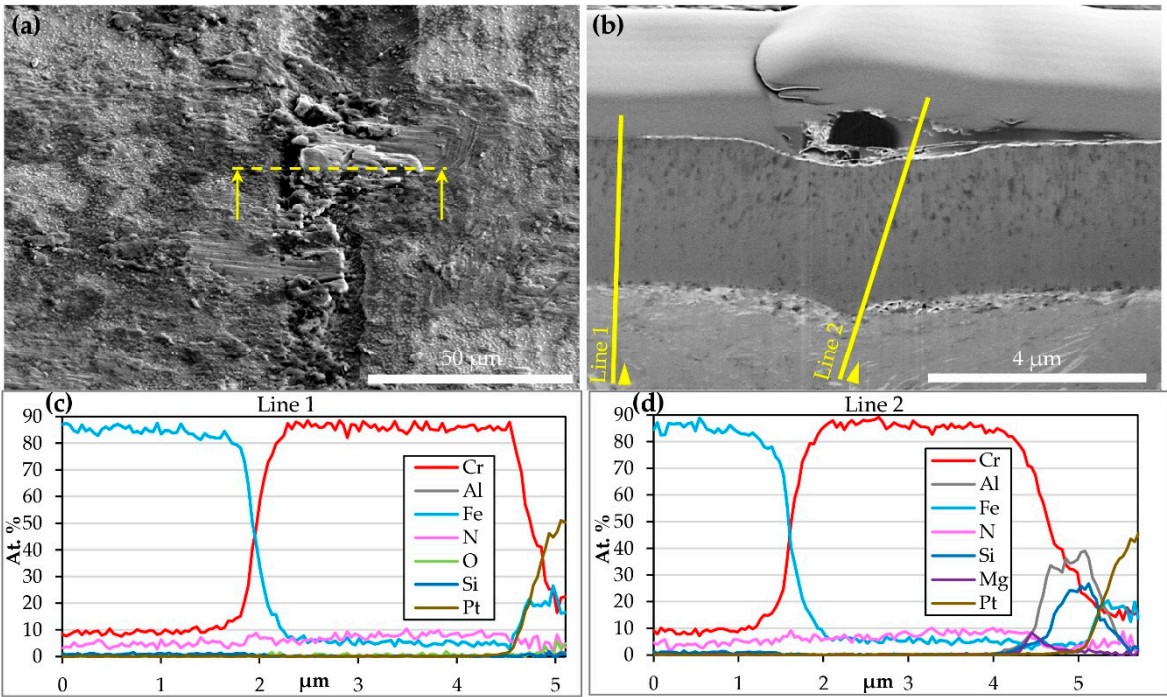

**Figure 4.** Focused ion beam (FIB) analysis of the CrN-R pin sample with aluminum built-up after casting solidification (CS) experiment, (**a**) secondary electron image, line and arrows indicate FIB milling line and the direction of cross section observation, (**b**) cross-sectional ion induced secondary electron image, (**c**) and (**d**) chemical composition obtained by EDS analysis conducted on cross section along lines drawn in the image (b).

Results of a cross-sectional EDS line analysis are presented in Figure 4c,d. The analysis was performed at two locations, at location without a built-up layer (line 1) and at location with the built-up layer (line 2). The chemical composition of CrN coating layer measured on two locations is similar. The Cr content is very high and almost constant, while the N content is low but also constant across the coating layer. Although a thin bright layer is observed on the top of the coating (Figure 4b), the expected increase of O content was not detected in neither of two tested locations. EDS lines presented in Figure 4d show that intermetallic phases inside the built-up layer consist of Al, Si, and Mg.

Results of FIB analysis performed on CrN-R sample subjected to DS 20 experiment are presented in Figure 5. The cross-sectional analysis was performed on a location where thin cast alloy built-up layer was present, Figure 5a. The CrN coating was not damaged mechanically, however there were changes in coating microstructure and phase constitution, Figure 5b. The bottom layer of CrN coating appears darker than the rest of the coating, which suggests his layer is of different chemical and/or phase composition. Crystallographic (ion channeling) contrast of the same region shown in Figure 5c, indicates that the bottom layer has large, elongated, crystals aligned perpendicular to the interface. On the top of CrN coating, a very thin surface layer can be seen, Figure 5b,c. The cast alloy built-up layer in the analyzed region is very thin and consists of phases (possibly intermetallic) with different crystallographic orientation.

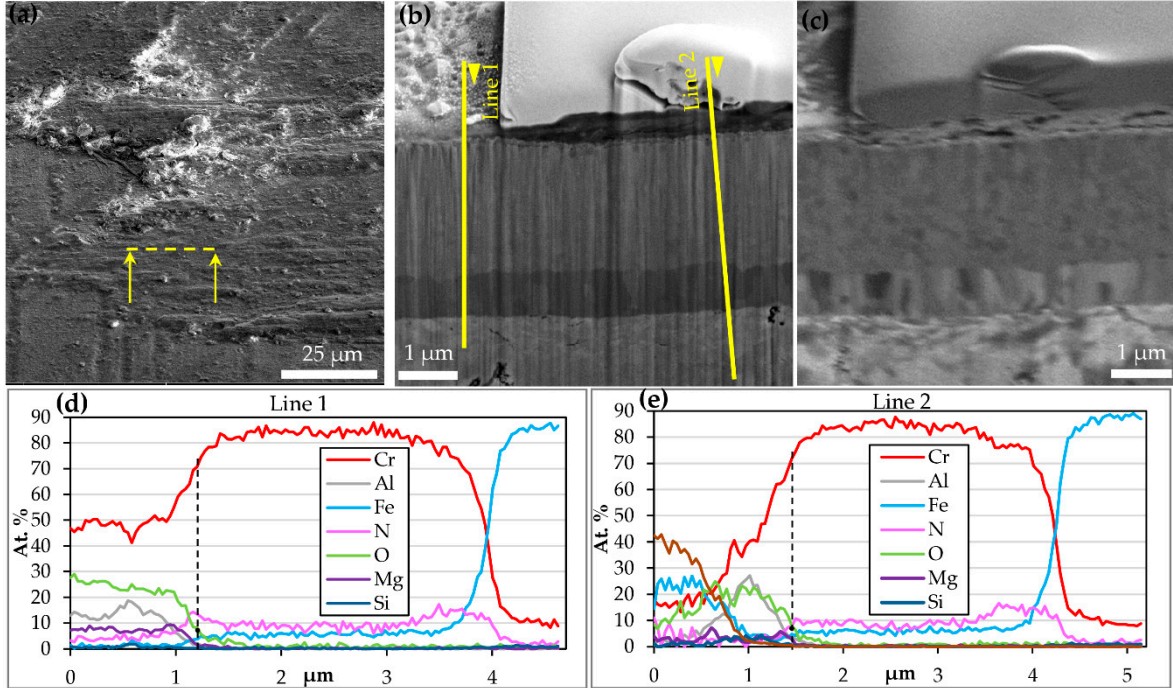

**Figure 5.** FIB analysis of the CrN-R pin sample with aluminum built-up after DS 20 experiment, (**a**) secondary electron image, line and arrows indicate FIB milling line and the direction of cross section observation, (**b**) cross-sectional secondary electron image, (**c**) cross-sectional ion induced secondary electron image, (**d**) and (**e**) chemical composition obtained by EDS analysis conducted on cross section along lines drawn in the image (**b**).

Results of EDS analysis performed on a cross section of CrN-R sample subjected to DS 20 experiment are presented in Figure 5d,e. The N content is a bit higher (lower Cr/N atomic ratio) in a top thin layer and substantially higher in a bottom thicker layer. The bottom layer of increased N content corresponds to the darker layer with elongated crystals, seen in Figure 5b,c. Increased values of O, Al, and Mg were detected in top regions of both CrN coating and built-up layer. It is worth to note that the line of O content does not follow the line of Al and Mg content. At certain depths (indicated by dashed lines in Figure 5d,e) the O content is fairly high, while the content of Al and Mg is low and insignificant. This means that the top of CrN layer is oxidized (Cr–O formed) and that above it a layer of Al and Mg oxides formed, out of constituents of the casting material.

### 3.4. ToF-SIMS Analysis of Samples after Ejection Tests

In order to determine more accurately the phases that constitute different layers observed in FIB images, ToF-SIMS analysis was engaged. Depth profiling was performed on pin locations which were not exposed to a cast alloy and on locations which were exposed to a cast alloy. ToF-SIMS depth profiles obtained for the most representative samples are presented in Figures 6 and 7. Signals of CrO$^-$, AlO$^-$ and CrN$^-$ secondary ions are presented, since they are of the most importance for the present study.

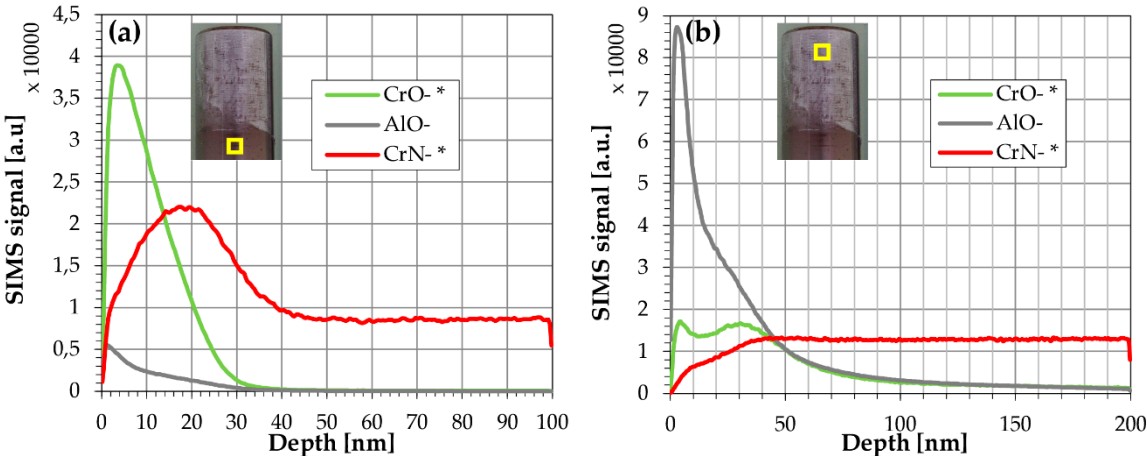

**Figure 6.** ToF-SIMS depth spectra of the CrN-R sample after CS experiment: (**a**) profile at the location not exposed to cast alloy; (**b**) profile at the location exposed to cast alloy, with built-up layer. The intensity of signals marked with * is multiplied five times. Figure inserts show the investigated pin surfaces and the locations of the depth profiling.

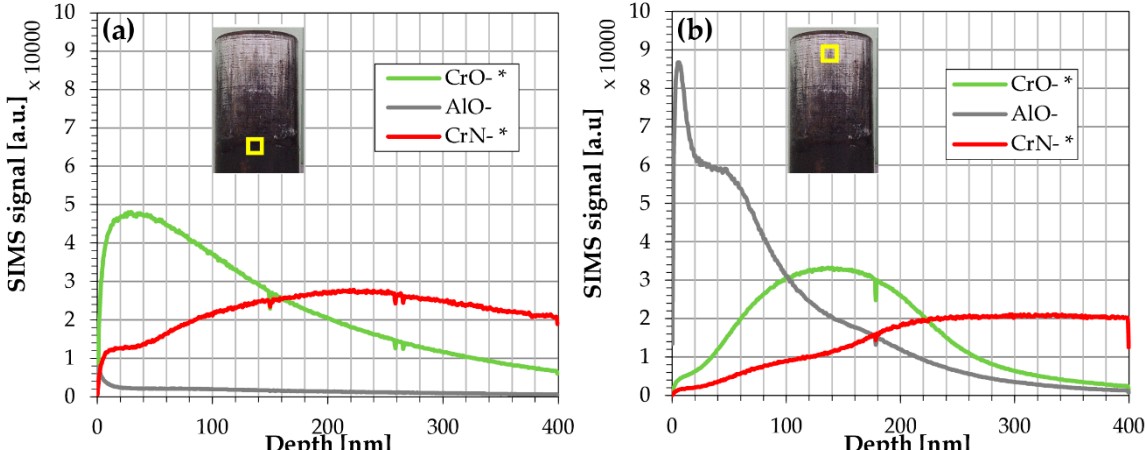

**Figure 7.** ToF-SIMS depth spectra of the CrN-R sample after DS 20 experiment: (**a**) profile at the location not exposed to cast alloy; (**b**) profile at the location exposed to cast alloy, with built-up layer. The intensity of signals marked with * is multiplied 5 times. Figure inserts show the investigated pin surfaces and the locations of the depth profiling.

A very thin surface layer of Cr–O (15–20 nm) is found at both analyzed location of CrN-R sample from CS experiment. This layer formed on the top of CrN coating, Figure 6. A peak of CrN⁻ near the surface region is not realistic, it is an artefact of the applied measuring technique. It formed due to the matrix effect caused by presence of an oxide [25]. At the location which was not exposed to a cast alloy a smaller quantity of Al–O was found in a thin surface layer, Figure 6a. On the other side, at the location which was exposed to a cast alloy a 15 nm thick layer with higher quantity of Al–O was detected, Figure 6b. It has to be noted that the thin layer with high Al–O content also has lower content of Cr–O.

Figure 7 shows ToF-SIMS depth profiles of a CrN-R sample subjected to the DS 20 experiment. A quite thick Cr–O layer (~150 nm), in which very small quantity of Al–O is identified, is present at the location which was not exposed to a cast alloy. In the top surface layer, the content of CrN is low, it increases with the depth as the content of Cr–O decreases. A top layer of Al–O with a low content of Cr–O and CrN is present at the location which was exposed to a cast alloy. Deeper in the layer, the content of Al–O decreases, while contents of Cr–O and CrN increases. Cr–O content reaches its maximum at the lower depth than CrN which means that Cr–O layer lies over the CrN. The thickness

of the Cr–O layer in this region is approximately 150 nm. As the content of Cr–O declines the content of CrN rises and reaches its maximum. Similar findings were obtained from ToF-SIMS analysis of samples subjected to DS 5 experiments.

### 3.5. Cross-sectional Analysis of Samples Subjected to Annealing Test

FIB cross-sectional analysis of the CrN sample annealed in air at 650 °C is presented in Figure 8. The analysis is performed at location where a nodular defect was present in the coating. In the middle of the coating microstructure is unchanged. Similar grain size and distribution are observed as in as-deposited CrN coating presented in Figure 2a. On the other side, microstructure of CrN top and bottom layer transformed during the annealing experiment. A thin bright layer formed in the top part of the coating with a layer consisting of larger elongated crystals beneath it, Figure 8a. A similar layer, consisting of large elongated crystals is present in the bottom part of CrN coating. Such crystals are comparable to those observed in the bottom part of CrN-R sample subjected to DS 20 experiment, presented in Figure 5. Results of a cross-sectional EDS line analysis are shown in Figure 8b. Note that these results are presented in the number of counts which means that they can be used for qualitative analysis. A top thin layer (~200 nm) has increased content of O. Considering that in this top layer the content of Cr matches with the content of O, this is probably Cr–O layer. The EDS analysis showed that both (top and bottom) layers with large elongated crystals have increased content of N and somewhat lower content of Cr.

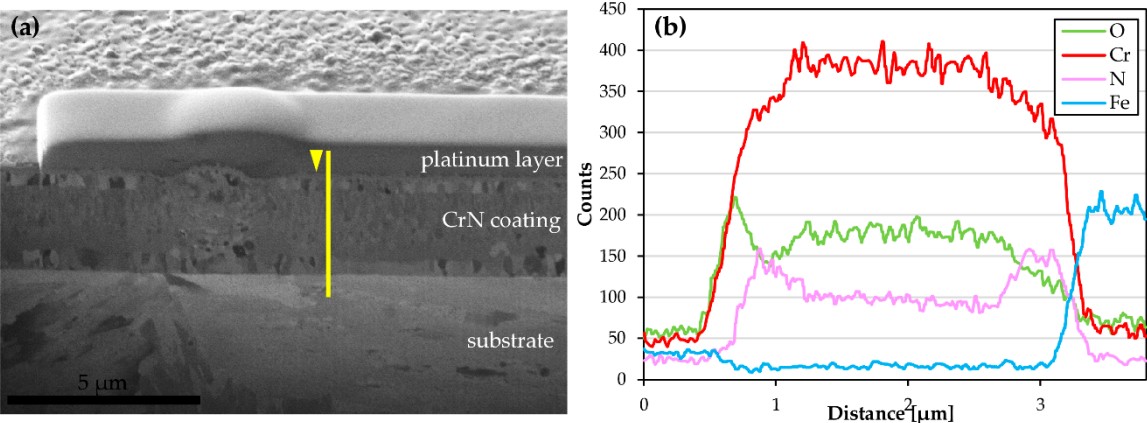

**Figure 8.** CrN coating in annealed state, (**a**) ion induced secondary electron image of CrN coating cross-section in the annealed state, (**b**) chemical composition obtained by EDS analysis along the line depicted in the image (**a**).

## 4. Discussion

### 4.1. Initial Coating Properties

Qualitative results of EDS analysis indicate that the investigated CrN coating has under-stoichiometric composition ($Cr_2N$). Its chemical composition is comparable to the chemical composition of CrN coating from previous studies, which were produced in similar conditions using the same deposition chamber [23,24]. High hardness for CrN coating is attributed to its dense, fine-grained microstructure and under-stoichiometric coating composition [26]. Besides high hardness, such a microstructure ensures a high cracking resistance. PVD coatings usually consist of columnar grains where cracks propagate easily between the grains [27,28]. However, when fine grains constitute the microstructure, as in our CrN, cracks have to bend around the grains which results in increased cracking resistance. High cracking resistance is of high importance because coatings applied on HPDC tools are subjected to thermal fatigue caused by alternating heating and cooling cycles inherent in the casting process [2]. Additionally, the underlying plasma nitrided layer has very important role in

the composite system. It has high hardness (1300 $HV_{0.1}$) and therefore it provides high load bearing capacity for CrN coating.

*4.2. Chemical Composition and Microstructure After Casting Experiments*

Compared to CS experiments, DS casting experiments significantly altered coating chemical composition and microstructure. The observed changes have specific impact on the behavior of coated samples during the ejection tests. Therefore, to explain the differences observed in the ejection tests, changes in coating chemical and microstructural properties have to be addressed first in detail.

CrN coating was not significantly changed after it was exposed to testing conditions in CS experiment. Microstructure was unchanged, while chemical composition was altered only in a very thin layer at the top of the coating. A thin (15–20 nm) Cr–O layer was formed. This layer should be $Cr_2O_3$ [26,29], which formed during the casting process when the pin surfaces were shortly exposed to temperatures up to ~600 °C.

It is known that $Cr_2O_3$ layer forms on CrN coating in these conditions [30]. ToF-SIMS analysis (Figure 6) shows that oxide layer forms more intensively at pin locations which were not exposed to a cast alloy, i.e., it forms more readily at locations which were exposed to air in die. A thin Al–O layer (detected only by ToF-SIMS), is the casting oxide scale which forms when molten aluminum alloy comes in contact with air. However, considering the standard Gibbs free energy of formation, from Ellingham diagram [31], the Al–O layer observed in this study could also form by reduction of the Cr–O top layer by Al from the casting alloy. The time that a liquid cast alloy stays in a contact with a pin sample is very short in CS experiment [21]. This explains why the Al–O layer on the cast alloy is not so thick. The changes observed in CrN layer after conducting the CS experiments are not detrimental to coating integrity. In the industrial use of the coating on HPDC tools, the observed changes would appear after the first few hundreds, or thousand, of casting cycles.

Two types of changes were observed in samples after the DS experiments. The first type involved formation of oxide top layer and its interaction with the cast alloy. The second involved changes in microstructure and chemical composition just beneath the top oxide layer and at the bottom of CrN coating layer. Oxidation of CrN coating occurred during preheating stage in which samples could reach temperatures above 650 °C. Formation of oxide at the top of the coating was also confirmed by EDS study conducted on the annealed samples, Figure 8. Oxide layer in CrN coatings forms due to outward diffusion of Cr and inward diffusion of O through the formed oxide layer [9,26,29]. For under-stoichiometric CrN ($Cr_2N$) coating, oxidation process forms a layer of $Cr_2O_3$ [26,29]. This layer suppresses the outward diffusion of N [26,29], as a consequence the Cr/N ratio right beneath the oxide layer is decreased, Figure 5. As a result, stoichiometric CrN forms beneath the oxide layer [26]. Such a layer is favorable for applications on HPDC tools because it oxidizes slower and forms denser microstructure than the initial under-stoichiometric CrN [9,26]. However, large elongated crystals present in the CrN layer beneath the oxide does not promise high mechanical properties.

Both EDS and ToF-SIMS analyses (Figures 5 and 7) showed that the cast alloy built-up layer on CrN coatings subjected to DS experiments consists of a relatively thick Al–O layer. This Al–O layer formed in a reaction of Al–Si–Cu cast alloy with the top Cr–O layer. This analysis showed that during DS experiments coating is subjected to diffusion wear by cast alloy, which should be classified as the coating metallurgical soldering mechanism. Thickness of Al–O layer depends on two factors, the first is the thickness of an initial Cr–O layer, and the second is the time a coated sample spends in contact with a molten alloy. Both are more pronounced in DS tests than in CS tests, i.e., thickness of Cr–O layer is larger and the exposure time to molten alloy is longer (~35 min, [32]), which explains why Al–O layer is thicker after DS tests. Upon casting solidification, the pin-casting contact is basically established between Cr–O and Al–O layers (Cr–O/Al–O pair).

On the bottom of CrN coating, an N-rich layer formed due to outward diffusion of N from the underlying nitrided steel, which is expected to occur at temperatures higher than 550 °C [33]. Due to the increased N content, a stoichiometric CrN was formed. Large, elongated grains aligned

perpendicularly to the interface constitute this layer, which is similar to microstructure observed in the layer present in the top area just beneath the oxide layer. If the coating would be exposed to similar casting conditions for a longer time, the two N-rich layers would eventually meet and the whole CrN coating would consist of large elongated grains. Although in a real industrial production die surfaces can be exposed to temperatures as high as 650 °C, since production cycles are short such N-rich layers cannot form in a single cycle or after smaller number of cycles. However, after exposing the coating material to high temperatures in tens of thousands of casting cycles N-rich layers are likely to occur.

Changes observed in annealed sample agree very well with the above discussed changes observed after DS tests. This implies that changes induced during DS tests appear as a consequence of heating and exposure to an oxidizing environment. The only difference in results obtained in these two experiments is the thickness of oxide and "nitrogen-rich" layers. Both layers were thinner in CrN coatings subjected to DS experiments. Oxide layer is thinner due to the shorter period of exposure to air and quite possibly due to the reduction of Cr–O by Al contained in a casting alloy.

### 4.3. Ejection Force

Although DS experiments were performed without repetition, results of the ejection test should be regarded as reliable. Sequence of steps performed in a DS experiment is almost equal to sequence performed during a CS experiment. Two experiments differ only in selection of parameters, such as preheating temperature and the delay of the solidification process, which were precisely controlled. Therefore, we postulated that the expected variation in the ejection force (coefficient of variation) in DS experiments should be similar to variation obtained for CS experiments, approximately 10% of the measured value [21]. Small variations in the ejection force obtained for different samples in DS 5 and DS 20 experiments (Figure 3) corroborate this statement.

In the next paragraphs we will discuss on values of the ejection force measured for different samples, in different testing conditions.

High values of the ejection force were measured in CS experiments. In CS experiment, the pin-casting contact is established between a Cr–O coating layer and a very thin layer of Al–O contained in a casting. Formation of these layers was discussed in detail in the previous subsection. In the initial stage of the ejection process, the Al–O layer is easily removed because the cast alloy is very soft, and as such it does not provide enough support for very thin Al–O layer. As a consequence, pin samples were mostly sliding against surface of Al–Si–Cu cast alloy. In rare publications on this topic, it is documented that such sliding occurs with a considerable friction, because aluminum alloy increases the adhesive wear component [34]. In the cited study, high friction coefficient of 0.5 was measured between Cr–O and Al in a pin-on-plate test [34]. Even higher friction coefficient (>1.4) was recorded in cross-cylinder tests [35,36]. Therefore, high values of the ejection force obtained in CS experiments are in agreement with high coefficient of friction characteristic for Cr–O/Al pair [34–36].

Values of the ejection force recorded in both DS 5 and DS 20 experiments were substantially lower than in CS experiment. In DS experiments the pin-casting contact is established between thick a Cr–O coating layer and a thick layer of Al–O formed on casting surface. Considering that these layers have substantial hardness, they did not wear off during the pin ejection process. Due to their high hardness [37] and high chemical inertness, these layers in tribological contact impede adhesive wear and high friction [34,38]. For example, for Cr–O/$Al_2O_3$ pair in a pin-on-plate test friction coefficient of 0.4 was measured, while in work [39] friction coefficient was between 0.35 and 0.4. Therefore, lower values of the ejection force recorded in DS experiments are attributed to lower coefficient of friction of materials in contact.

Surface roughness strongly affected the ejection force in CS experiment, where the ejection force increased with the decreased roughness. A detailed study on the mechanical soldering mechanisms responsible for such a trend was provided in one of our previous studies [3]. On the other hand, an almost equal ejection force was measured in DS experiments for samples of different roughness. These observations along with the above provided discussion suggest that in DS experiments effects of

surface chemistry are more dominant than the effects of surface topography, i.e., that metallurgical mechanisms are more dominant than mechanical mechanisms.

*4.4. Coating Changes in Delayed Solidification Experiments and Implications on its Performance in HPDC Process*

The observed transformation of CrN duplex layer has several drawbacks. Formation of large elongated (columnar) grains in stoichiometric CrN layers might be detrimental for application on HPDC tools. Such layers have lower mechanical properties [28], lower resistance to crack propagation, and are prone to intergranular sliding during severe mechanical loading [27,28]. Additionally, a columnar grain structure creates diffusion paths between grains which enhance diffusion of O [9].

Results of this study unambiguously show that formation of oxide layer on CrN coating is highly beneficial for reduction of ejection force and cast alloy soldering. This is in agreement with findings published in [1,40], in which the application of $Cr_2O_3$ as a working layer of HPDC tools is highly recommended. However, diffusion wear of Cr–O layer observed in this study can induce negative effects in a long run of HPDC production. The issue is the cyclic exchange of oxidation and diffusion wear processes, which significantly contribute to the overall wear of a coating. If the coating diffusion wear would be slowed down, or brought under the control, the benefits of Cr–O application would prevail. This might be achieved either by application of thick Cr–O working layers or by nanolayer coating design. We propose a coating of a nanolayer design where nanolayers of Cr–O and O-diffusion barrier material would be alternatively deposited. Besides low ejection forces and reduced soldering effect, such a design would suppress the oxygen diffusion out of the coating layer. Application of coatings with Cr–O ("sacrificial") layer allows design of die-cores without drafts. In this way the technological limitation of the casting design is overcome, which is extremely beneficial in HPDC technology.

**5. Conclusions**

Soldering performance of duplex CrN coating was evaluated by modified ejection test with the delayed casting solidification (DS). Solidification was delayed for 5 min and 20 min. Obtained results were compared to the results from our previous study [3], in which the same coating was evaluated in the conventional solidification (CS) experiment.

In both experiments with the delayed casting solidification the ejection force, required for separation of CrN coated pin samples and Al-alloy castings, was considerably reduced compared to the conventional solidification tests. Depending on sample roughness, the ejection force was reduced from 20% to 50% of the value obtained in conventional solidification experiments.

In order to understand these differences and to study the wear mechanisms acting in different experimental setups, thorough characterization of sample surfaces covered with built-up layers was conducted. It was found that in both conventional and delayed solidification experiments Cr–O formed at the coating side while Al–O formed at the casting side. Cr–O oxides formed when CrN coating was exposed to air at high temperatures, while Al–O formed in Al–Si–Cu alloy by reduction of Cr–O by Al from the alloy. Since the time CrN coating was exposed to both air and liquid cast alloy was longer in the DS experiments, both Cr–O and Al–O layers were substantially thicker in the DS tests. Considering that reduction of Cr–O occurred by diffusion of O into the cast alloy, this kind of wear should be regarded as coating metallurgical soldering. Such a wear mechanism has not been reported in the literature from the field, so far.

Besides oxidation, microstructural and compositional changes were found in top and bottom layers of CrN coating after DS experiments. Two N-rich layers formed, one just beneath the Cr–O oxide, and the other just above the nitrided steel. The first layer formed as a consequence of CrN oxidation, and the second as a consequence of N outward diffusion from the underlying nitrided layer. In both layers elongated grain structure was found.

During ejection of coated pins from castings formed in CS experiments, at the beginning Cr–O layer was in the contact with a very thin Al–O layer. This thin Al–O layer was easily removed, and the sliding mainly occurred between Cr–O and Al–Si–Cu alloy. High adhesion between these materials resulted in high ejection force. On the other hand, during ejection in DS experiments a thick Al–O casting scale inhibited the contact of Al–Si–Cu alloy with a thick Cr–O layer. As a consequence, soldering and friction between a coated pin and a casting was reduced and lower ejection force was recorded.

It was found that pin samples with different surface roughness in DS experiments exhibited approximately the same values of the ejection force. This is contrary to CS experiments where roughness played important role. Differences arise because the pin-casting material pair formed in DS experiments (Cr–O/Al–O) greatly reduces adhesion and galling which both increases with the decrease in surface roughness. These observations show that in DS experiments the effect, of surface chemistry is more dominant than the effect of surface roughness.

This study showed that modification of ejection test by introduction of the delayed casting solidification is appropriate approach for introduction of severe soldering and corrosion conditions which are required for appropriate evaluation of metallurgical soldering performance of coating materials. Such a test configuration allowed us to recognize several mechanisms of CrN coating soldering wear, which were not addressed in the literature so far.

In order to confirm the findings presented herein, while excluding the negative effects of DS experiments, for future investigations we suggest performing the CS experiment with previously oxidized CrN coating. The other important point is determination of the intensity of coatings diffusion wear in contact with aluminum alloy castings. In such a way it can be determined whether the CrN oxidation prior to application on HPDC tools is beneficial. Additionally, the soldering performance and wear of $Al_2O_3$ coating and/or $Al_2O_3$-forming (i.e., TiAlN, TiAlSiN) coatings should be compared with $Cr_2O_3$ coating and/or $Cr_2O_3$-forming coatings (i.e., CrN, CrAlN).

**Author Contributions:** Conceptualization, P.T. and L.K.; methodology, P.T., L.K., A.M. and B.Š.; validation, L.K. and A.M.; investigation, P.T., L.K., J.K. and A.D.; resources, B.Š. and J.K.; writing—original draft preparation, P.T.; writing—review and editing, P.T., L.K., A.M. and B.Š.; visualization, A.M.; supervision, B.Š.; project administration, B.Š.; funding acquisition, B.Š. All authors have read and agreed to the published version of the manuscript.

**Funding:** This research was funded by Serbian-Slovenian bilateral project (2018–2019) grant 48. This work was also funded by the Slovenian Research Agency (program P2-0082) and European Regional Development Funds (CENN Nanocenter, OP13.1.1.2.02.006).

**Acknowledgments:** Special thanks to Peter Panjan, Institute "Jožef Stefan" (Ljubljana, Slovenia), for help in samples characterization and fruitful discussions about the results obtained in this study. Additionally, we are very grateful to Miha Čekada, Institute "Jožef Stefan" (Ljubljana, Slovenia) for all the help and for providing the resources needed for the investigation. Authors also gratefully acknowledge Termometal d.o.o. (Ada, Serbia) for samples and tools production.

**Conflicts of Interest:** The authors declare no conflict of interest. The funders had no role in the design of the study; in the collection, analyses, or interpretation of data; in the writing of the manuscript, or in the decision to publish the results.

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
