# Peer review of "Metallurgical Soldering of Duplex CrN Coating in Contact with Aluminum Alloy"

_coatings, doi:10.3390/coatings10030303_

Reviewer 1 Report
1. This article explores the use of a coated tool for high pressure die casting. In general, the article is of certain interest to the potential reader, but needs a number of additions and changes. Among other things, you should additionally check the English grammar.
Thank you for your comments and suggestions. We checked the English grammar ones again, and we made changes throughout the manuscript.
2. Abstract must be radically changed. The Abstract does not need an introductory part or descriptions of how widely this or that technology is used. The usual structure of the abstract is as follows: "Statement of the problem - solution methods - main results." An abstract is not part of the article, so there is no point in explaining the abbreviations in it. An abstract is a separate and self-sufficient text, on the basis of which a potential reader must understand whether this article is interesting to him or not.
Yes, you are right about the abstract structure. However, the problem investigated in the manuscript is quite a specific topic, and it is not so frequently addressed in literature of surface engineering. Therefore, in the first sentence we just wanted to inform the reader about the topic addressed in the manuscript and to gently introduce it to the problem. We completely changed the first sentence to emphasize the problem analyzed in the manuscript (rows 13-15). In the second sentence (rows 15 and 16) we stated the problem more specifically. In the third and the following sentences we presented the experiment and the main results.
Abbreviations are used in the abstract to reduce the number of words in it, because the space for the abstract is quite limited, 200 words. They were mainly used to shorten the terms conventional and delayed solidification. Additionally, in the guide for manuscript preparation (https://www.mdpi.com/journal/coatings/instructions) it is written that abbreviations should be used in the first place of their appearance in the abstract, main text or figures... And, therefore we acted accordingly.
3. The list of keywords is too long. Typically, a semicolon rather than a comma is used to separate keywords.
According to the guide for manuscript preparation, Journal Coatings allow the use of up to ten keywords. However, we shortened the phrases in the keywords and reduced the number of keywords to nine. In the lists of keywords (row 28 and 29) commas are replaced by semicolons.
4. The introduction should primarily justify the choice of research direction. However, in this case there is completely no justification for choosing a CrN coating. Why exactly this coverage? There are dozens of alternative options ... I would like to see an overview of the coatings used for these purposes, indicating the pros and cons of various coating compositions and, ultimately, justifying the choice of CrN. I must note that this coating is not distinguished by either better wear resistance or better heat resistance, so its choice is not obvious.
Thank you very much for this remark we noticed a deficiency of our manuscript and we made changes. Let us elaborate. The focus (direction) of this work was on metallurgical soldering of PVD coatings which is usually overlooked in the investigations from the field. Additionally, we wanted to demonstrate that the modified ejection test simulates the metallurgical soldering effects of PVD coatings. In the fifth paragraph of introduction section we explained the needs for such kinds of investigations. We have chosen the CrN coating because of few reasons. CrN has fairly high hardness, high oxidation and corrosion resistance, low stress and also a fair thermal stability. Owing to such properties CrN coating is still a standard coating for protection of tools used in hot processing of aluminum alloys (forming, deep drawing, HPCD...). Nowadays HPDC tools are also protected by chromium based ternary and quaternary coatings produced with different designs i.e. CrAlN, AlCrN, CrAlSiN, Cr2O3, but also Al2O3 coating. Considering that Cr – N constitute most of these coatings detailed information on its metallurgical soldering behavior would help in understanding the behavior of more complex Cr based coatings. Additionally, experimental evaluation of the soldering performance of a coating that does not contain aluminum makes the chemical characterization after soldering tests much easier. Therefore, we considered this coating as an appropriate model for soldering evaluation. Our intention was not to show that CrN is the best choice for soldering reduction. However, on that example we wanted to test soldering resistance in more severe environment, and we found that coatings can suffer from metallurgical soldering in aluminum alloy, which is often overlooked in the literature.
In order to convey this information more clearly, we made the following changes in the article. We changed the manuscript title to: Metallurgical soldering of duplex CrN coating in contact with aluminum alloy; in introduction section we added the text about other coatings applied on HPDC tools (rows 53-58 ); to better convey the manuscript main focus we modified the sentences in the fifth paragraph of the introduction section (rows 94-95, 97); to explain the choice of CrN coating we added two sentences to the last paragraph of introduction section (rows 119-122); in conclusions sections we added a sentence (rows 546 and 547).
5. The two-stage process (nitriding in one installation + coating in another installation) is not very rational. There are technologies nitriding + deposition of the coating in one process. In this case, in particular, repeated surface preparation (polishing) is not required.
Yes, you are right, there are such processes. However, for the presented investigation the way that duplex composite was obtained is not important, it is just a model of material (layer) which is the most appropriate for protection of HPDC tools. In the investigated case multiple polishing procedures were applied to produce samples of different roughness.
6. It is desirable to present an image of the coating structure with good resolution and scale. In Figure 2 and 4, the coating looks single-layer, in Figure 5 two layers are clearly visible, and in Figure 8 three layers. What are these layers? They need to be described at the beginning. A larger magnification and better contrast are needed to study the crystal structure of the coating layers.
Yes, the investigated CrN coating is a single-layer coating whose initial cross section is presented in Figure 2a and explained in rows 231-233 (we added the term single-layer). Figure 4b shows the coating cross-section after CS experiments, and sentences in rows 270-272 explain that the coating layer did not change during this test, which is also discussed in 2nd paragraph of section 4.2 sentences from (rows 391-393). Figure 5b,c and 8a presents the coatings cross-sections after DS experiment and annealing experiments, respectively. These experiments induced considerable changes of the coating layer, which are for Figure 5 b and c detailly explained in the sentences from rows 286-291 and discussed in the 4th and 6th paragraph of section 4.2.; and for Figure 8a it is explained in rows 358-365 and discussed in 4th,6th and 7th paragraph of section 4.2.
7. The outer coating layer shown in Figure 8 may also be a consequence of spinodal decomposition (see doi:10.1016/j.vacuum.2004.08.020, Acta Materialia 61 (2013) 7534–7542, DOI 10.1016/j.wear.2019.203069)
It is not clear to which layer does the reviewer imply. After annealing, there is an outer oxide layer on top of CrN coating, and there is a layer with altered composition and microstructure in the upper portion of CrN coating. We will comment on both layers.
Oxide layer: We provided exhaustive discussion on formation of the outer oxide layer (rows 407-415), which was based on studies which detailly addressed oxidation of CrN ((https://doi.org/10.1016/0040-6090(96)08663-4; https://doi.org/10.1016/j.surfcoat.2012.01.033.; https://doi.org/10.1016/j.tsf.2013.01.031.; https://doi.org/10.1016/0257-8972(94)90139-2.). Spinodal decomposition is certainly not a mechanism leading to oxide formation.
Altered CrN layer: Spinodal decomposition is indeed mechanism observed in TiAlN, and TiAlSiN coatings. It is possible, but highly unlikely to happen in CrAlN because of the low de-mixing energy of Cr1-xAlxN system where solid solution is highly stable (https://doi.org/10.1016/j.actamat.2007.04.029). After spinodal decomposition TiAlN and CrAlN coatings consist of c-TiN + c-AlN and c-CrN + c-AlN, respectively. However, there is no spinodal decomposition in CrN system, therefore, this mechanism does not lead to changes observed in the upper portion of the investigated CrN coating.
8. “Qualitative results of EDS analysis indicate that the investigated CrN coating has under stoichiometric composition (Cr2N)” - It is advisable to justify this in more detail.
In section 3.1 the sentences from rows 225-237 explain the CrN initial composition and clearly justify such statement.
9. “fine grains constitute the microstructure” - Where does this come from? Where is the crystal structure of this coating studied? How is a small grain size achieved?
In the results section, the images of CrN coating cross section (Figure 2a) indicate the coating fine-grained compact microstructure. This result is explained in the sentences from rows 231-233. The microstructure is studied by ion induced secondary electron imaging which gives a crystallographic contrast which arises due to differences in ion channeling in phases with diverse crystalline structures and orientations. This is also explained in the sentences from rows 274, 275, 295, 296. The way the microstructure and small grain size in CrN coating were achieved is out of the subject of this study and is therefore not addressed and discussed.
10. The "underlaying plasma nitrided layer" was practically not investigated in this work - thickness, hardness, chemical composition.
The properties of the plasma nitriding layer produced in presented investigations are given in the first paragraph of Materials and layers properties section 3.1, in the sentences from rows 227-229.
11.The phrase "most probably" is not very suitable for a scientific article!
Yes, we agree. We made changes in the sentence from row 393, and used references which confirm that statement
Reviewer 2 Report
The manufacturing of non-ferrous alloys is realized mainly by high pressure die casting (HPDC). The wear of the die tool surfaces and the obtainment of the desired quality surface of the casted pieces is still an issue. The paper treats the afore mentioned aspect, proposing the deposition of CrN duplex coating on HPDC tools.
I appreciate the thoroughly experimental research carried out by the authors, the results being interpreted and validated with the aid of previously obtained ones by the same authors, or using results reported in literature.
Unfortunately, the proposed solution seems not to be an efficient one at this moment, as some diffusion wear was reported. Anyway, the study of the problem of coating metallurgical soldering seems to be an original contribution of this work. Also, a possible solution to prevent the diffusion wear was proposed: the use of previously oxidized CrN coatings.
Thank you very much for highlighting the quality, contribution and significance of our work presented in the paper.
I consider that the paper can be improved, and I have some suggestions.
1. In Introduction section should be mention some details about the previous undertaken research in this field, not only general phrases enumerating the bulk references.
Yes, you are right, eight sentences are added to the 4th paragraph of introduction section (rows 63-77). These sentences give details about the investigations previously performed in the field and the most important findings obtained in these investigations.
2. Even if most of the research was previously published by the authors, some clarifying figures reproduced from those articles would be welcome.
Answering to the previous remark required a considerable extension of the introduction section, which is now two pages in the length. Considering that incorporation of figures into the introduction section would additionally increase its length, we think that such a change is not convenient. Additionally, we do not find any of figures from our previous works appropriate to additionally clarify statements provided in the introduction section.
3. The presentation of a general testing plan in a table is requested. This should present the next aspects: tested samples and die tools, thermal treatments, roughness etc.
We agree. The requested table is added in the section 2.1 in row 165. Accordingly, the table in which the roughness parameters were previously presented in section 3.1 is deleted.
4. A diagram for the applied treatment procedures of each experiment (quenching, annealing, tempering) is recommended.
One sentence is added to the section 2.1 (rows 134-136) which explains the quenching and tempering procedure that was applied in this investigation. We avoided the use of a diagram for this purpose, since the whole Materials and Methods section is already quite long.
5. Economically speaking, a delayed solidification at 700 0C for 20 minutes is cheaper than supporting some wear on die tool surfaces?
We are not sure that we understood the question correctly. A delayed solidification method was developed just for experimental purposes. The delayed solidification process extends the period during which cast alloy is in the contact with the investigated pin sample material. During the preheating process, a pin is heated up to 600֯C which can cause material oxidation. In our case CrN oxidized and formed a top layer of Cr-O. The pins with an oxide layer displayed very good ejection performance. Considering that HPDC technology is a mass production process in which castings are rapidly solidified, application of the delayed solidification is not economical at all. In our work we proposed that CrN coated pins should be oxidized before use in real HPDC tool, which is economically justified.
6. Why a quantitative EDS of the deposited coatings was not realized?
The EDS line analysis on the as-deposited coating was performed solely to characterize the uniformity of chemical composition across the coating thickness. Since the investigated coating was produced by a regularly used deposition procedure, its chemical composition is very well known. The same CrN deposition procedure was used in previous works, we referred to one of them for more details on the quantitative chemical composition.